# ATP activates bestrophin ion channels through direct interaction

Yu Zhang[1], Alec Kittredge[1], Nancy Ward[1], Changyi Ji[1], Shoudeng Chen[2] & Tingting Yang[1]

Human Bestrophin1 (hBest1) is a $Ca^{2+}$-activated $Cl^-$ channel in retinal pigment epithelium (RPE) essential for retina physiology, and its mutation results in retinal degenerative diseases that have no available treatments. Here, we discover that hBest1's channel activity in human RPE is significantly enhanced by adenosine triphosphate (ATP) in a dose-dependent manner. We further demonstrate a direct interaction between ATP and bestrophins, and map the ATP-binding motif on hBest1 to an intracellular loop adjacent to the channel activation gate. Importantly, a disease-causing mutation of hBest1 located within the ATP-binding motif, p. I201T, diminishes ATP-dependent activation of the channel in patient-derived RPE, while the corresponding mutants in bestrophin homologs display defective ATP binding and a conformational change in the ATP-binding motif. Taken together, our results identify ATP as a critical activator of bestrophins, and reveal the molecular mechanism of an hBest1 patient-specific mutation.

[1] Department of Pharmacology and Physiology, University of Rochester, School of Medicine and Dentistry, Rochester, NY 14642, USA. [2] Molecular Imaging Center, Department of Experimental Medicine, The Fifth Affiliated Hospital of Sun Yat-sen University, Zhuhai, Guangzhou 519000, China. Correspondence and requests for materials should be addressed to T.Y. (email: tingting_yang@urmc.rochester.edu)

Human Bestrophin1 (hBest1) is encoded by the *BEST1* gene and mainly expressed in retinal pigment epithelium (RPE)[1–3]. The genetic mutation of *BEST1* causes at least five retinal degenerative diseases[2–9], notably Best vitelliform macular dystrophy (Best disease). Clinical phenotypes of *BEST1*-mutated patients include retinal detachment and progressive vision loss; no treatment is currently available. Belonging to a protein family consisting of four members (Bestrophin1–4), hBest1 has been functionally identified as a $Ca^{2+}$-activated $Cl^-$ channel (CaCC) in heterologous studies[10–15]. Importantly, the majority of hBest1 disease-causing mutations are point mutations likely associated with channel dysfunction[11,16]. Thus, understanding how hBest1 activity is regulated in the retina and is impacted by the numerous mutations holds tremendous value for both biological and biomedical perspectives.

As a CaCC, hBest1 showed a high $Ca^{2+}$ sensitivity with half maximal effective concentration ($EC_{50}$) at ~150 nM when heterologously expressed in human embryonic kidney (HEK)293 cells[12,17], consistent with results from several other bestrophin homologs in mouse (mBest2), human (hBest4) and *Xenopus laevis* (xBest2)[15,18–20]. Purified chicken Bestrophin1 (cBest1) displayed an even higher $Ca^{2+}$ sensitivity in bilayer ($EC_{50}$ 17 nM)[21]. However, if $Ca^{2+}$ binding is sufficient for activation in physiological conditions, one would expect that native bestrophin channels remain constantly activated, as the physiological basal level of free cytosolic $Ca^{2+}$ concentration ($[Ca^{2+}]_i$) is typically ~100 nM. Moreover, $Ca^{2+}$-independent activation has been reported for several bestrophin homologs, such as a purified bacterial bestrophin from *Klebsiella pneumoniae* (KpBest) measured in lipid bilayer[13], and human Bestrophin2 (hBest2) and *Drosophila* Bestrophin1 (dBest1) heterologously expressed in HEK293 cells[11]. These functional results suggest that bestrophins either remain constantly open or have additional activator(s) besides $Ca^{2+}$ under physiological conditions.

Structurally, if $Ca^{2+}$ is sufficient to activate hBest1, one would expect $Ca^{2+}$-bound channels to likely be in an open state. Although the structure of hBest1 is still unavailable, the crystal structures of KpBest and cBest1 have been solved at 2.3 and 2.9 Å resolution, respectively[13,14]. However, neither structure is in an open state, despite that cBest1 was solved in complex with $Ca^{2+}$. There are two possibilities why an activator-bound channel is not open: firstly, bestrophins may have other unidentified coactivator(s) and secondly, $Ca^{2+}$ may mediate both channel activation and inactivation/rundown. However, this latter scenario is hard to reconcile with the unchanged cBest1 structure in the presence of different concentrations of $Ca^{2+}$[14]. Notably, KpBest and cBest1 have very similar structures, although the former shares a much lower sequence identity (14%) with hBest1 compared to the latter (74%).

Here, we discover a direct interaction between adenosine triphosphate (ATP) and bestrophins, which is essential for channel activation. The highly conserved structures of bestrophin homologs and the low sequence identity shared between KpBest and hBest1 provide a unique opportunity to systematically determine critical residues on the channel. Using purified KpBest mutant proteins as a probing tool, we map a critical ATP-binding motif adjacent to a conserved activation gate in the channel ion conducting pathway. Importantly, hBest1-mediated endogenous $Ca^{2+}$-activated $Cl^-$ current in human induced pluripotent stem cell (iPSC) derived RPE (iPSC-RPE) displays ATP-dependent activation, while a disease-causing mutation I201T within the ATP-binding motif displays impaired ATP-dependent activation in patient-derived iPSC-RPE. Moreover, structural analysis of the KpBest L177T mutant (equivalent to hBest1 I201T) provides further insights into ATP binding and channel activation. Taken together, our results uncover ATP as an interacting activator of bestrophins and the molecular mechanism of a *BEST1* patient-specific mutation.

## Results

**ATP activates and directly interacts with KpBest.** A critical clue about the activation of bestrophins was found in bilayer experiments with purified KpBest, as ATP, but not $Ca^{2+}$, significantly increases channel open probability ($P_o$) in a dose-dependent manner (Fig. 1a, b, Supplementary Figure 1a). A plot of $P_o$ as a function of ATP concentration ([ATP]) displayed robust ATP-dependent activation with $EC_{50}$ of ATP at 485 μM (Fig. 1b). These results indicate that ATP is an essential activator of KpBest. Since no other protein was involved in the bilayer experiment system (Supplementary Figure 1b, c), we speculated that ATP directly interacts with and activates KpBest.

To directly assess the physical interaction between KpBest and ATP, the in vitro binding affinities of purified KpBest to ATP analogs (ATP, ADP, AMP, and ATPγS) were examined by microscale thermophoresis (MST). We found that KpBest has a higher affinity to ATP ($K_d = 254$ μM) than to ADP ($K_d = 1.3$ mM), no affinity to AMP, and the highest affinity to nonhydrolysable ATPγS ($K_d = 16$ μM) among all tested analogs (Fig. 1c, d, Supplementary Figure 1d). These results demonstrate a direct interaction between KpBest and ATP without the requirement of ATP hydrolysis.

**ATP stimulates hBest1-mediated $Cl^-$ current in human RPE.** The role of ATP as an interacting activator of KpBest prompted us to test its regulatory involvement with hBest1, which functions as an essential CaCC to mediate $Ca^{2+}$-dependent $Cl^-$ current in human RPE[16]. To test whether ATP is an activator of hBest1 under physiological conditions, we examined the influence of ATP on endogenous $Ca^{2+}$-dependent $Cl^-$ current in human RPE cells.

iPSCs were derived from hBest1 WT donor skin cells and then differentiated to iPSC-RPE[16]. As previously reported, hBest1 was localized on the plasma membrane of iPSC-RPE (Fig. 2a). $Ca^{2+}$-dependent $Cl^-$ currents in iPSC-RPE were examined by whole-cell patch clamp across a range of intracellular free [ATP] ($[ATP]_i$), while $[Ca^{2+}]_i$ was maintained at 0.6 μM (Fig. 2b–d). Remarkably, currents were $98 \pm 32$ pA pF$^{-1}$ when $[ATP]_i$ was 0, and increased in amplitude as $[ATP]_i$ was raised from 100 μM to 2 mM, peaking at $301 \pm 92$ pA pF$^{-1}$ with 2 and 10 mM $[ATP]_i$ (Fig. 2c, d, Supplementary Table 1). A plot of peak current (evoked with a 100 mV step pulse) as a function of $[ATP]_i$ displayed robust ATP-dependent activation with the $EC_{50}$ at 677 μM (Fig. 2c). Similar ATP-dependent $Cl^-$ current profiles were recorded in iPSC-RPEs derived from two distinct clonal iPSCs of the same donor (Supplementary Table 1). To examine the involvement of phosphorylation-mediated regulation, we substituted ATP with nonhydrolyzable ATPγS. The $Ca^{2+}$-dependent $Cl^-$ current in iPSC-RPE was enhanced in response to ATPγS (Fig. 2d), suggesting that phosphorylation is not required for the stimulatory effect of ATP.

Next, we asked if ATP by itself can activate hBest1 in iPSC-RPE. Tiny currents (<5 pA pF$^{-1}$) were recorded without $Ca^{2+}$ even when $[ATP]_i$ was at a saturated concentration of 10 mM, and current amplitudes increased as $[Ca^{2+}]_i$ was raised to 0.6 μM while $[ATP]_i$ was held at 10 mM (Fig. 2d, Supplementary Figure 2a), indicating the essential requirement of $Ca^{2+}$ and the insufficiency of ATP alone in hBest1 activation. Moreover, the measured currents were inhibited by niflumic acid (NFA), a $Cl^-$ channel blocker, confirming that these are indeed $Cl^-$ currents (Fig. 2d). Taken together, our results identified ATP as a coactivator of hBest1: $Ca^{2+}$ is necessary for channel activation,

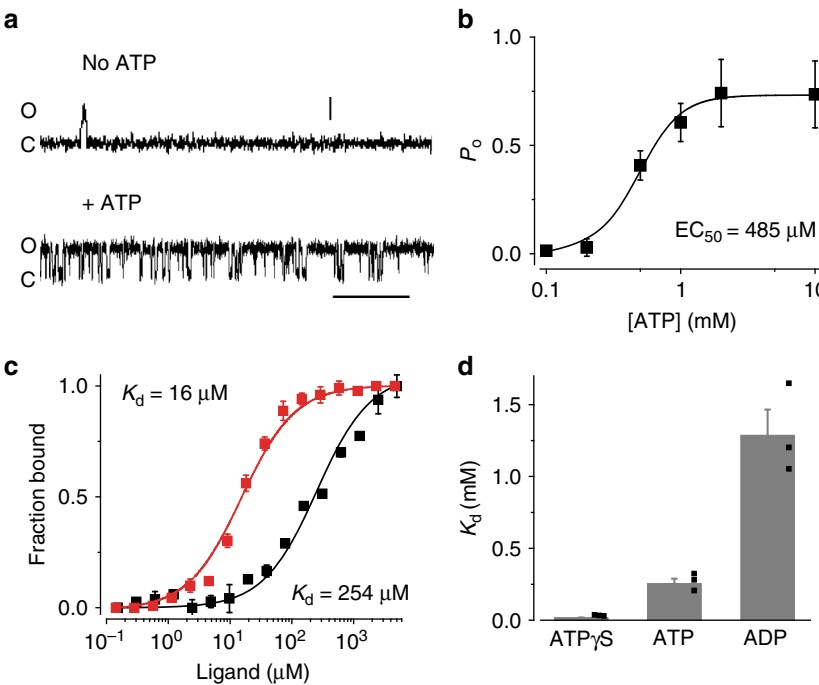

**Fig. 1** The influence of ATP on KpBest and the interaction between them. **a** Current traces of single KpBest channels recorded from planar lipid bilayers at 80 mV in the absence (top) and presence (bottom) of 2 mM ATP (Scale bar, 3.5 pA, 250 ms). **b** The open probability of the KpBest channel with different concentrations of ATP. $n = 3$ for each point. The plot was fitted to the Hill equation. **c** The MST binding curves of KpBest to ATP (black) and ATPγS (red). Protein fraction bound vs. ligand, $n = 3$ for each point. **d** Bar chart showing the binding affinities of KpBest to ATP analogs. $n = 3$ for each bar. All error bars in this figure represent standard error of the mean (s.e.m.)

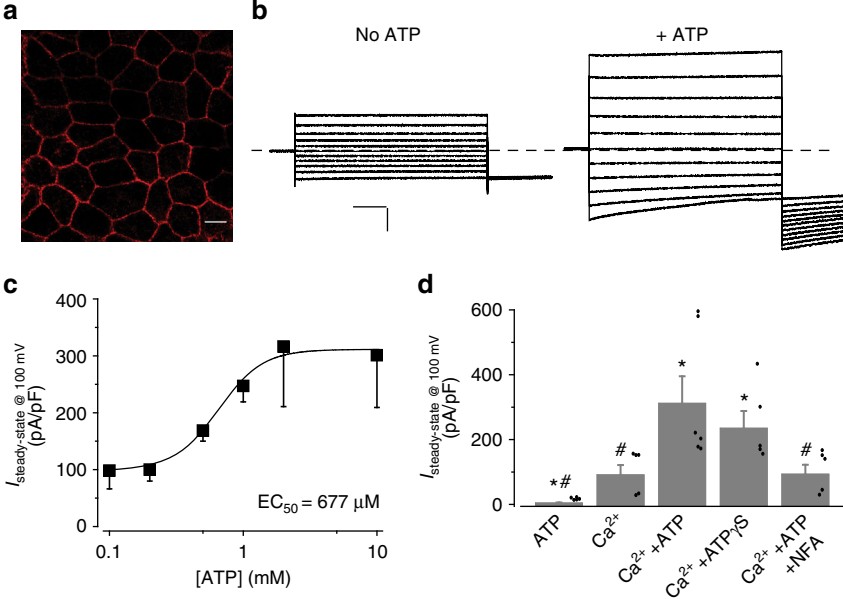

**Fig. 2** The influence of ATP on hBest1-mediated $Ca^{2+}$-dependent $Cl^-$ current in human RPE cells. **a** Confocal image showing plasma membrane localization of hBest1 in human RPE (Scale bar, 10 μm). **b** Representative current traces recorded from a WT iPSC-RPE in the absence (left) or presence (right) of 2 mM ATP (Scale bar, 1 nA, 125 ms). **c** ATP-dependent activation of surface currents in WT iPSC-RPE. Steady-state current density recorded at +100 mV plotted vs. free $[ATP]_i$, $n = 5$–6 for each point. The plot was fitted to the Hill equation. **d** Bar chart showing the steady-state current amplitudes in the presence of ATP (10 mM) without $Ca^{2+}$, $Ca^{2+}$ (0.6 μM) without ATP, $Ca^{2+}$ (0.6 μM) + ATP (10 mM), $Ca^{2+}$ (0.6 μM) + ATPγS (10 mM), and $Ca^{2+}$ (0.6 μM) + ATP (10 mM) + NFA (100 μM) in WT iPSC-RPE, $n = 5$–6 for each bar. $^{*\#}P < 0.05$ compared to currents with $Ca^{2+}$ and $Ca^{2+}$ + ATP, respectively, using one-way ANOVA and Bonferroni post hoc analyses. All error bars in this figure represent s.e.m.

and ATP significantly stimulates channel activity by about threefold. Notably, the currents in iPSC-RPE ran up after patch break regardless of the presence of ATP (Supplementary Figure 2b, c).

**ATP interacts with and activates a mammalian bestrophin.** As ATP activates KpBest through direct interaction, we speculated that the same mechanism is at play for hBest1. To circumvent the unavailability of purified hBest1 protein, we obtained purified

bovine Bestrophin2 (bBest2) containing amino acids 1–406 of the protein (Supplementary Figure 3a). It has been shown that the highly conserved N-terminal region of hBest1 (residues 1–390, 61% sequence identity with bBest2; Fig. 3a) is sufficient to conduct CaCC activity[12], while the C-terminal region of hBest1 (residues 391–585) is much less conserved among different bestrophin homologs/paralogs, and predicted to be unstructured.

Strikingly, purified bBest2 showed very similar binding affinities to ATP analogs in MST as those of KpBest, with $K_d$s to ATPγS, ATP and ADP at 10 μm, 560 μm, and 3.0 mM, respectively (Fig. 3b, c, Supplementary Figure 3b). Consistently, the channel activity of bBest2 transiently expressed in HEK293 was significantly stimulated by ATP in whole-cell patch clamp (Fig. 3d). Taken together, our results strongly suggest that ATP binding is an evolutionarily conserved mechanism for the activation of bestrophin channels.

**Mapping ATP interacting residues in KpBest.** No conventional ATP-binding pocket, usually composed of an alpha-helix and beta-sheet mix[22], was found in either KpBest or cBest1. Therefore, we attempted to map the ATP interacting site(s) on bestrophins using the following rationale: firstly, as both KpBest and bBest2 showed similar binding affinities with ATP analogs, and all three tested bestrophin homologs including hBest1 showed ATP-dependent activation, it is reasonable to predict ATP-binding site(s) at conserved regions among these three species; secondly, as ATP is localized at the intracellular side of the plasma membrane (in the internal solution during patch clamp recordings), ATP-binding site(s) should be located within the major cytoplasmic regions of bestrophin channels; thirdly, as neither KpBest nor cBest1 contains any beta-sheet in the structure, ATP binding is likely mediated (at least partly) by loop(s), which offer structural flexibility to accommodate ATP and the associated conformational changes. Based on these three criteria, four candidate ATP-binding motifs (loops 1–4) on the intracellular loops of bestrophins were identified from an alignment of KpBest, hBest1, and bBest2 (Figs. 3a and 4a). Importantly, these motifs contain hot spots of disease-causing mutations, as all but one (hBest1 G199) of the identical residues among the three species in loops 1–3 have reported patient-specific mutations, and three residues have more than one mutation per amino acid position. The enrichment of mutations suggests that the candidate ATP-binding motifs contain key regulatory site(s) for the channel function, providing a validation of our reasoning.

To further narrow down which candidate motif(s) are critical for ATP binding, we comprehensively examined the influence of alanine substitution of the evolutionarily identical residues in each motif on KpBest and hBest1 (Fig. 4a). For each channel, the four mutants after alanine substitution in individual candidate motif were named A1–A4, respectively (Supplementary Figure 4a). KpBest A1–A4 mutants were all purified with a similar size exclusion profile to the WT (Supplementary Figure 4b), suggesting that the overall integrity of the channel is still retained after alanine substitution.

We first measured single channel properties of purified KpBest mutants in bilayer. KpBest A1 and A3 showed no ion conductance in the absence or presence of 2 mM ATP in bilayer (Fig. 4b, c), indicating a complete loss of channel function. KpBest A4 behaved the same as KpBest WT in bilayer (Fig. 4b, c, Supplementary Figure 4d), suggesting the noninvolvement of loop 4 in ATP binding. Strikingly, KpBest A2 was functional in bilayer, but displayed similar open probabilities in the absence or presence of 2 mM ATP (Fig. 4b, c, Supplementary Figure 4c), suggesting that loop 2 is specifically critical for ATP binding and ATP-dependent activation.

As ATPγS has the highest binding affinity among all ATP analogs, we then measured the affinity of ATPγS to purified KpBest mutants in MST. KpBest A4 showed a similar affinity to ATPγS ($K_d$ = 20 μM) as that of KpBest WT (16 μM) (Fig. 4d), consistent with our bilayer results that loop 4 is not involved in ATP binding. By contrast, KpBest A1, A2, and A3 all showed no significant ATPγS interaction (Supplementary Figure 4e, f). As among those three only KpBest A2 displayed channel activity in bilayer, we concluded that loop 2 is specifically involved in ATP binding and ATP-dependent activation in KpBest.

**Loop 2 on hBest1 is critical for ATP-dependent activation.** We then asked if loop 2 in hBest1, corresponding to residues 199–203, is indeed critical for ATP-dependent activation of the channel in human RPE. However, iPSC-RPEs carrying endogenous hBest1 A1–A4 mutants do not exist. Previously, we reported that the patient-specific recessive mutation hBest1 P274R is a null mutation due to structural disruption[16], and the complete loss of $Ca^{2+}$-dependent $Cl^-$ current in the hBest1 P274R iPSC-RPE was rescued by virus-mediated supplementation of WT hBest1[16]. Importantly, endogenous hBest1 P274R could not be co-immunoprecipitated with virally expressed WT hBest1 or any of the A1–A4 mutants, suggesting that the P274R mutant cannot interfere with the assembly of the pentameric WT/A1–A4 channels (Supplementary Figure 5). Thus, we reasoned that the P274R iPSC-RPE could be used as a blank background mimicking *BEST1* knockout, and the hBest1 WT and mutant channels could be virally expressed for testing their ATP-dependent activities.

Consistent with our previous results, no currents were recorded in P274R iPSC-RPE in the absence or presence of 2 mM ATP (Fig. 5a, b). By contrast, robust ATP-dependent $Cl^-$ currents were recorded in P274R iPSC-RPE complemented with WT hBest1-GFP expressing from a BacMam baculoviral vector (Fig. 5a, b), validating this system for studying ATP-dependency of hBest1.

The hBest1 A1–A4 mutant channels were individually expressed from a BacMam virus in P274R iPSC-RPE, and measured for their functions by whole-cell patch clamp. Consistent with bilayer results from KpBest, hBest1 A1 and A3 showed no current in the absence or presence of 2 mM ATP (Fig. 5c), indicating loss of channel function. hBest1 A4 displayed similar current amplitudes as that of hBest1 WT with or without ATP (Fig. 5c), confirming that loop 4 is not involved in ATP-dependent activation. Remarkably, hBest1 A2 conducted $Cl^-$ current irresponsive to ATP (Fig. 5c), suggesting that loop 2 is critical for ATP binding. These results are also consistent with the fact that patient mutations are reported in loops 1–3, but not loop 4.

**A patient mutation in loop 2 impairs hBest1 activation.** To test if endogenous hBest1 mutations in the mapped ATP-binding motif impair channel ATP-dependence, we examined $Cl^-$ current in iPSC-RPE derived from the skin fibroblasts of a patient donor carrying an hBest1 I201T mutation, which does not affect the overall hBest1 expression level[16]. $Cl^-$ current in I201T iPSC-RPE was not enhanced by increasing ATP from 0 to 2 mM in the internal solution (Fig. 6a, b, Supplementary Table 1), in sharp contrast to the threefold increase of $Cl^-$ current in BEST1 WT iPSC-RPE under the same conditions (Fig. 2c, d, Supplementary Table 1). Similar ATP-independent $Cl^-$ current profiles were recorded in iPSC-RPEs derived from two distinct clonal iPSCs of the same patient donor (Supplementary Table 1). Consistently, the corresponding KpBest L177T mutant showed a 10 times lower affinity to ATPγS compared to that of KpBest WT in MST

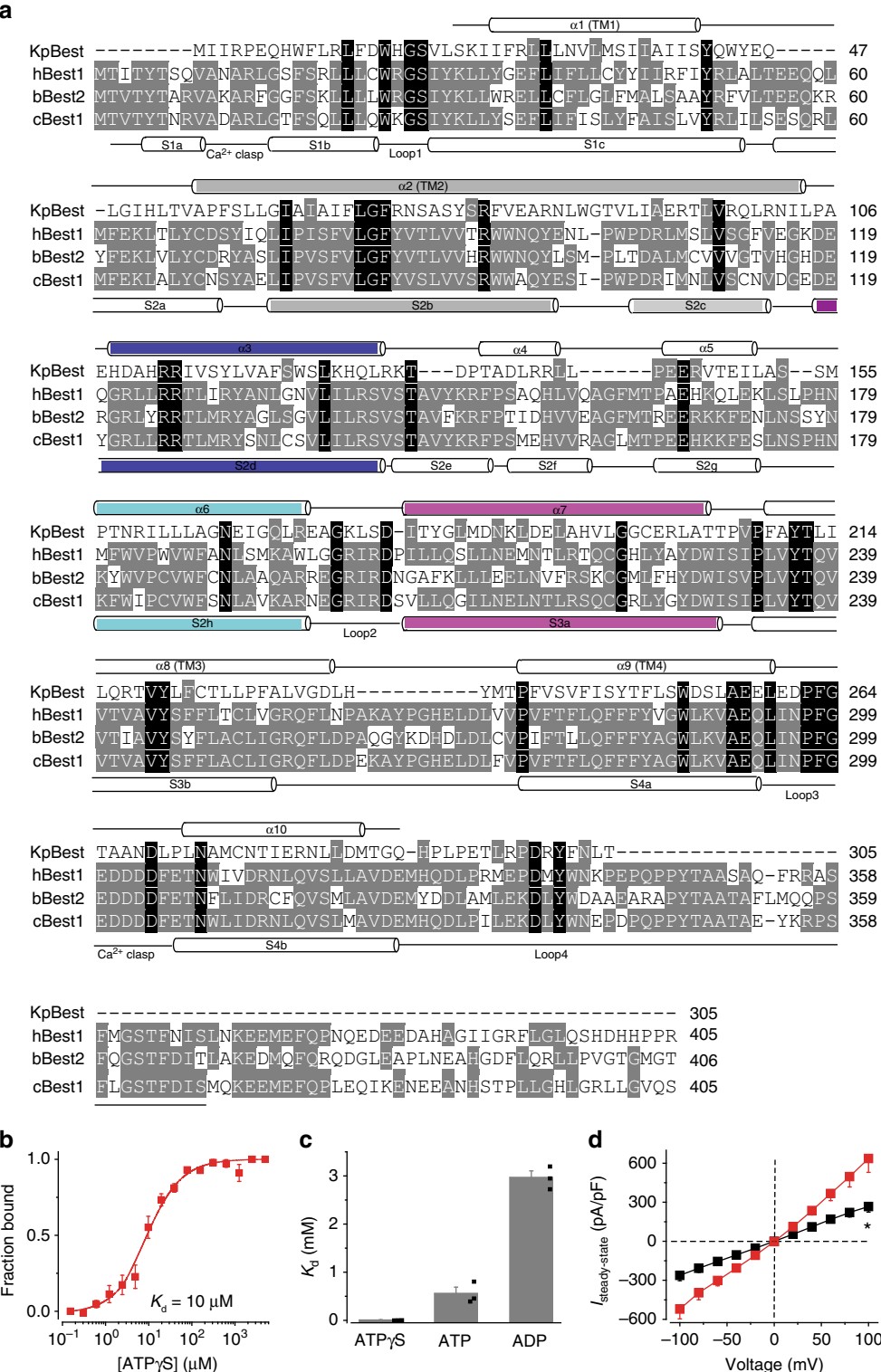

**Fig. 3** ATP interacts with and activates bBest2. **a** Structure-based sequence alignment of KpBest, hBest1, bBest2, and cBest1. The KpBest structure is used to restrict sequence gaps to interhelical segments. Black background, identical residues in all four species; gray background, identical residues in two or three species. The secondary structures of KpBest and cBest1 are labeled above and below the sequences, respectively. The four loops (1–4) potentially involved in ATP binding are labeled below the cBest1 secondary structure. Critical helices potentially involved in channel activation are highlighted in the same colors as those in Fig. 7. **b** The MST binding curve of bBest2 to ATPγS. Protein fraction bound vs. [ATPγS], $n = 3$ for each point. **c** Bar chart showing the binding affinities of bBest2 to ATP analogs. $n = 3$ for each bar. **d** Population steady-state current–voltage relationships of bBest2 transiently expressed in HEK293 cells without (black) or with (red) ATP (10 mM), $n = 9$–13 for each point. $^{*}P < 0.05$ compared to cells in the presence of ATP, using two-tailed unpaired Student $t$ test. All error bars in this figure represent s.e.m.

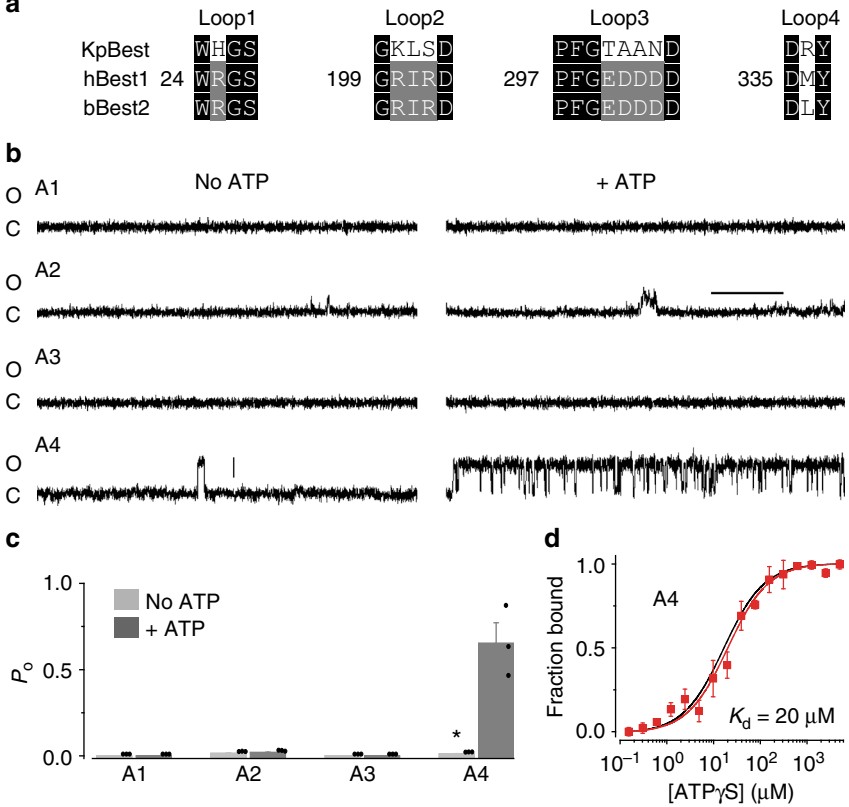

**Fig. 4** Mapping ATP-binding motif(s) in KpBest. **a** Candidate ATP-binding motifs in KpBest, hBest1, and bBest2. Black background, identical residues in all three sequences; gray background, identical residues in two sequences. Numbers indicate the position of the first residue in each motif on hBest1. **b** Current traces of single KpBest mutant channels recorded from planar lipid bilayers at 80 mV in the absence or presence of 2 mM ATP (Scale bar, 3.5 pA, 250 ms). **c** Bar chart showing the open probability of KpBest mutant channels in the absence or presence of 2 mM ATP. $n = 3$ for each bar. $^*P < 0.05$ compared to currents from the same channels in the presence of ATP, using two-tailed unpaired Student $t$ test. **d** The MST binding curve of KpBest A4 to ATPγS (red). Protein fraction bound vs. [ATPγS], $n = 3$ for each point. The binding curve of WT KpBest to ATPγS (black) is shown for comparison. All error bars in this figure represent s.e.m.

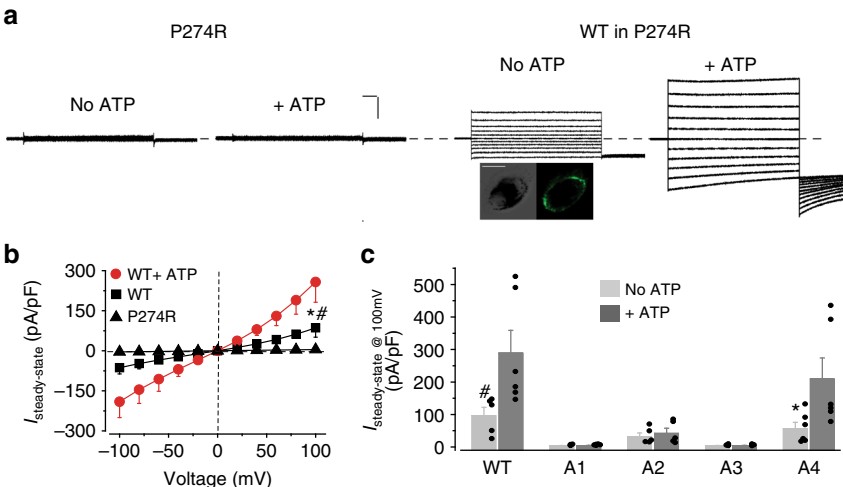

**Fig. 5** The role of candidate motifs on hBest1 in ATP-dependent activation. **a** Left, representative current traces recorded from hBest1 P274R iPSC-RPE. Right, representative current traces in hBest1 P274R iPSC-RPE complemented with WT hBest1-GFP (Scale bar, 1.5 nA, 100 ms). Insert, confocal images showing expression of WT hBest1-GFP in rescued P274R iPSC-RPE (Scale bar, 10 µm). **b** Population steady-state current–voltage relationships in P274R iPSC-RPE, and in P274R iPSC-RPE complemented with WT hBest1-GFP in the absence or presence of 2 mM ATP, $n = 5$–6 for each point. $^{*\#}P < 0.05$ compared to uninfected cells and complemented cells in presence of ATP, respectively, using two-tailed unpaired Student $t$ test. **c** Bar chart showing the steady-state current amplitudes of P274R iPSC-RPE complemented with WT or mutant hBest1 channels, $n = 5$–6 for each bar. $^{\#*}P < 0.05$ compared to currents from the same set of cells in the presence of 2 mM ATP, respectively, using two-tailed unpaired Student $t$ test. All error bars in this figure represent s.e.m.

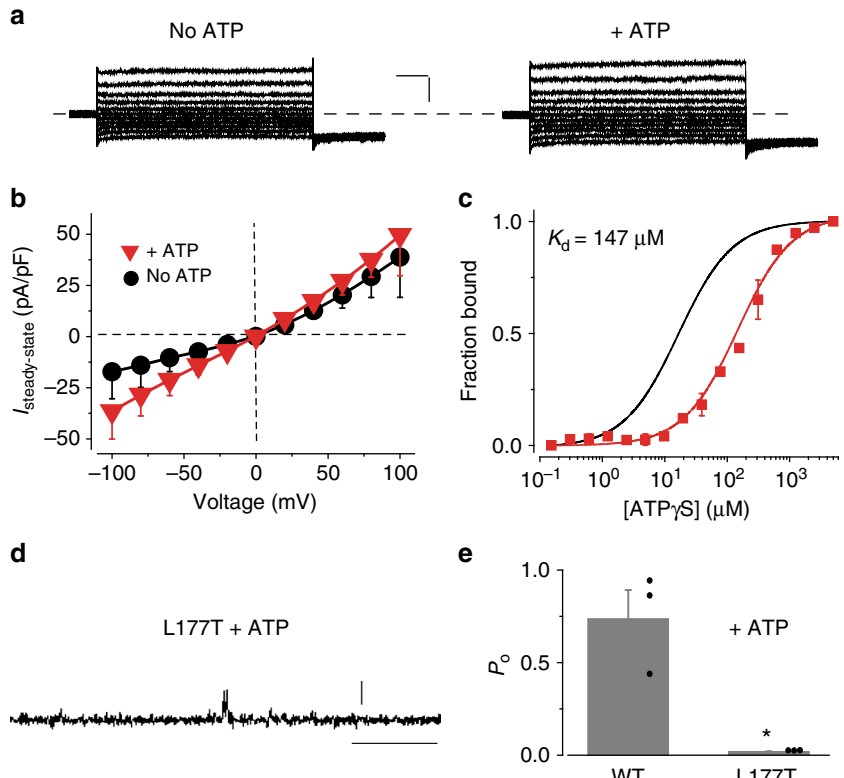

**Fig. 6** The influence of a patient-specific mutation on ATP-dependent activation. **a** Representative current traces recorded from hBest1 I201T iPSC-RPE in the absence or presence of 2 mM ATP (Scale bar, 300 pA, 100 ms). **b** Population steady-state current–voltage relationships in hBest1 I201T iPSC-RPE in the absence or presence of 2 mM ATP; $n = 5$–6 for each point. **c** The MST binding curves of KpBest L177T (red) to ATPγS. Protein fraction bound vs. [ATPγS], $n = 3$ for each point. The binding curve of WT KpBest to ATPγS (black) is shown for comparison. **d** Representative current trace of single KpBest L177T mutant channel recorded from planar lipid bilayers at 80 mV in the presence of 2 mM ATP (Scale bar, 2 pA, 250 ms). **e** Bar chart showing the open probability of KpBest WT and L177T channels in the presence of 2 mM ATP, $n = 3$ for each bar. $^{*}P < 0.05$ compared to the open probability of WT KpBest in the presence of ATP, using two-tailed unpaired Student $t$ test. All error bars in this figure represent s.e.m.

(Fig. 6c), and was not responsive to 2 mM ATP in bilayer (Fig. 6d–e). Moreover, the equivalent bBest2 I201T mutant was not stimulated by 10 mM ATP in transiently transfected HEK293 cells (Supplementary Figure 3c), and displayed no affinity to ATPγS in MST (Supplementary Figure 3d). These results suggest that the patient-specific hBest1 I201T mutation in the mapped ATP-binding motif causes defects in ATP binding and ATP-dependent activation, providing a disease-causing mechanism.

**Structural bases of loop 2 in ATP binding and activation**. We next sought structural mechanisms of the ATP-dependent activation. As the structure of hBest1 has not yet been solved, we constructed a three-dimensional human homology model based on the highly conserved KpBest and cBest1 structures (Fig. 7a, b). The pentameric hBest1 homology model displays a flower vase-shaped ion permeation pathway with two conserved hydrophobic permeation restrictions: a neck (I76, F80, and F84) and a putative activation/permeation gate (I205) (Fig. 7b)[13,14,16]. The ATP-binding motif (residues 199–203) is located on an intracellular loop (loop 2) adjacent to the channel activation gate (I205) (Fig. 7b–d), suggesting a potential role of this ATP-binding loop as a critical communication site for channel gating. Notably, the location and position of this loop within the channel structure are well persevered among species (Fig. 7c, d).

The ATP-binding motif in hBest1 contains two conserved arginine residues (R200 and R202), which are spatially close to two additional conserved arginine residues (R125 and R126, on helix S2d), providing the needed positive charge to form a complex with negatively charged ATP (Fig. 7c). Moreover, the hosting loop 2 not only connects helices S2h and S3a, but also interacts with helix S2d through a salt bridge between R200 and D118 (Fig. 7c), suggesting that the binding of ATP could cause significant conformational changes on these helices (e.g., by disrupting the salt bridge between R200 and D118). Importantly, as the activation gate I205 is on the N-terminus of helix S3a, ATP binding may cause structural alterations that affect the gate, leading to channel opening.

MST results from the equivalent KpBest and bBest2 mutants suggest that hBest1 I201T is deficient for ATP binding. As I201 is surrounded by evolutionarily conserved hydrophobic residues V114, A195, L207, and L210 (Fig. 7d, Supplementary Figure 6a), the substitution of a hydrophobic isoleucine to a polar threonine may impair the structure of the putative ATP-binding pocket by weakening the local hydrophobic interactions.

To gain more insight to the structural impact of the hBest1 I201T patient mutation, we revisited the previously solved crystal structure of the corresponding KpBest L177T mutant (Fig. 7d)[16]. The structure of KpBest L177T is very similar to that of WT KpBest, with all atom alignment root-mean-square deviation (RMSD) at 0.4 Å. However, superposition of the

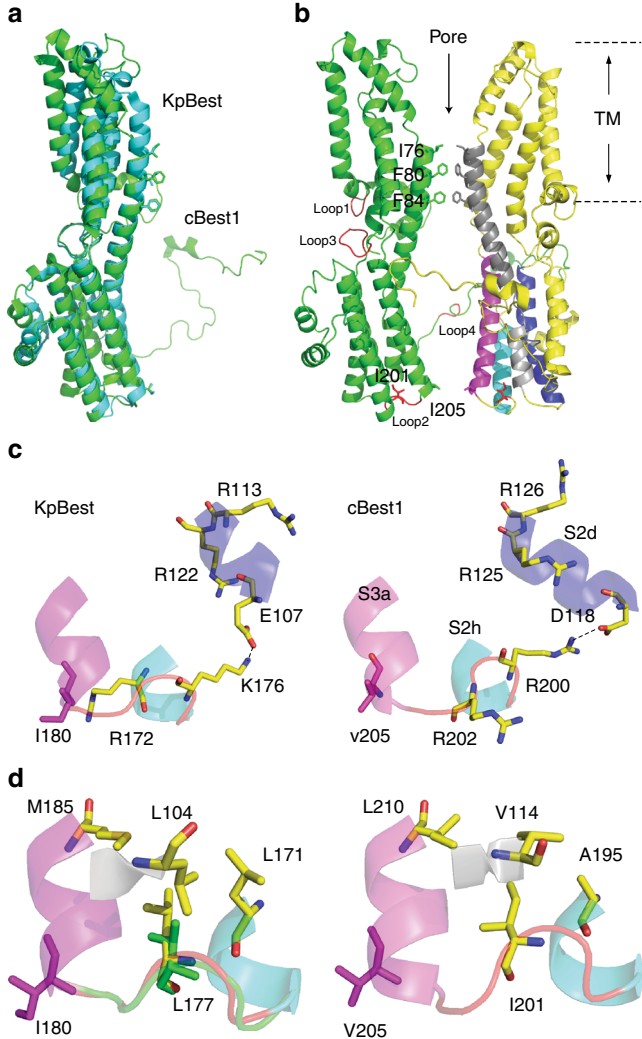

**Fig. 7** Structural analysis of bestrophin channels. **a** Structure alignment of KpBest (blue) and cBest1 (green) as shown by superposition of their protomers. **b** Ribbon diagram of two oppositely facing (144°) protomers of an hBest1 pentamer is shown with the extracellular side on the top. The side chains of I201 are in red. Loops 1–4 on the left protomer are in red. Helices surrounding the ATP-binding loop on the right protomer are labeled in the same colors as those in (c, d), Fig. 3A and Figure S5 for comparison. **c** Visualization of the ATP-binding loop (red), and critical residues potentially involved in ATP binding. Left, KpBest; right, cBest1. **d** Visualization of the ATP-binding loop (red for WT and green for the L177T mutant) and the surrounding hydrophobic residues. Left, KpBest; right, cBest1

KpBest L177T mutant with WT showed an obvious shift of the ATP-binding loop away from the hydrophobic region (Fig. 7d, Supplementary Figure 6b). Thus, the conformation of the ATP-binding loop is altered by a disease-causing mutation within it, providing a structural explanation for the defects in ATP binding and the subsequent channel activation.

## Discussion

Here, we discovered an evolutionarily conserved ATP-dependent activation mechanism of bestrophin channels by studying bestrophins from three different species—*Klebsiella pneumoniae*, bovine, and human, with multidisciplinary approaches including MST, lipid bilayer, patch clamp, and structural analysis. We further mapped an ATP-binding motif (residues 199–203 on

hBest1) located on an intracellular loop next to the channel activation gate (I205 on hBest1), providing a structural basis for the communication between ATP binding and channel gating. Importantly, we found that a patient-derived mutation (hBest1 I201T) located within the ATP-binding loop impairs ATP binding and ATP-dependent activation. Therefore, our findings not only provide insights to bestrophin channel properties, but also have direct implications in associated human diseases.

Consistent with previous studies identifying hBest1 as a CaCC, we showed that $Ca^{2+}$ is absolutely required for hBest1 activation, because no current was recorded in the absence of $Ca^{2+}$, even with saturated ATP (10 mM). With a median level of $Ca^{2+}$ (0.6 μM), ATP significantly enhanced hBest1 current amplitude by threefold. These results support a model in which $Ca^{2+}$ is an essential constitutive activator of hBest1, while ATP serves as a coactivator to stimulate the activation of $Ca^{2+}$-bound hBest1. The next question is how $Ca^{2+}$ and ATP cooperate to activate hBest1? Two possible scenarios are: (1) $Ca^{2+}$ can only partially activate hBest1, while ATP is required for the full activation and (2) $Ca^{2+}$ can fully activate hBest1, while ATP shifts the $Ca^{2+}$ sensitivity of hBest1 in favor of channel opening. Existing evidence supports the first model: structurally, the $Ca^{2+}$-bound cBest1 is not in an open state;[14] functionally, I201T impaired ATP-dependent activation without shifting the $Ca^{2+}$-sensitivity of $Cl^-$ current in patient iPSC-RPE[16]. We propose a two-step activation model for bestrophin channels (Supplementary Figure 7): without $Ca^{2+}$ and ATP, the channel is in a closed state; the $Ca^{2+}$-bound channel is in a partially open state; the channel bound by both $Ca^{2+}$ and ATP is in a fully open state. Further investigation will be needed by varying $[ATP]_i$ at different $[Ca^{2+}]_i$s to obtain a complete $Ca^{2+}$-ATP codependent activation profile of hBest1.

It remains unclear how ATP binding activates bestrophin channels. Notably, in hBest1 and cBest1, the evolutionarily conserved hydrophobic residues around I201 are located on three different helices—S2c, S2h, and S3a (Fig. 7d, Supplementary Figure 6a), two of which are critically involved in forming the ion conducting pathway: S3a hosts the channel gate, while S2c connects through a very short loop (four residues) to the TM helix S2b, which contains the three neck residues (Figs. 3a and 7a). Interestingly, the shortly spaced helices S2b and S2c in hBest1/cBest1 are merged into one long TM helix α2 in KpBest (Fig. 3a), underlying the tight connection between S2b and S2c, and potentiating the mutual influence from one helix to the other. Therefore, we speculate that ATP binding mediated by the ATP-binding loop triggers a conformational change through the three structurally adjacent helices to broaden both the gate and the neck, which are the two restrictions in the ion conducting pathways of KpBest and cBest1 (Fig. 7a, b, Supplementary Table 2). Nevertheless, obtaining a ligand-bound hBest1 structure will provide a clear picture of this important mechanism.

Nonhydrolysable ATPγS shows the strongest binding affinity to KpBest and bBest2 in MST experiments (Figs. 1 and 3), and stimulates hBest1 in RPE (Fig. 2d), suggesting that neither ATP hydrolysis nor phosphorylation is required for the interaction and subsequent channel activation. Moreover, the $K_d$s measured in MST, representing the affinities between ATP and the channels, were in a similar range of the $EC_{50}$s in bilayer and patch clamp recordings, which represent the activation of the channels, despite the different setups of these experiments. For instance, the $K_d$ and $EC_{50}$ to ATP for KpBest were 254 and 485 μM, respectively; the $K_d$ for bBest2 was 560 μM, while the $EC_{50}$ for hBest1 was 677 μM. As the free ATP concentration in cytoplasm is approximately 400–600 μM[23], our results suggest that physiological regulation of ATP levels in the cell will have significant effects on the activities of bestrophin channels.

Although hBest1 is characterized as a CaCC, $Ca^{2+}$-independent activation has been reported for other bestrophins, such as KpBest in lipid bilayer, and hBest2 and dBest1 heterologously expressed in HEK293 cells[11]. As bBest2 shares 85% and 91% sequence identity to hBest2 in the full length and N-terminal TM region, respectively, it is very possible that ATP directly interacts with and activates hBest2. Moreover, endogenous dBest1–4 are sensitive to intracellular ATP: in whole-cell patch clamp experiments, addition of ATP to the internal solution augmented the activation of $Ca^{2+}$-dependent $Cl^-$ current by twofold, and also accelerated the current run-up; in excised patches, bestrophin channels were activated in only a small number of patches exposed to a high concentration of $Ca^{2+}$ without ATP within 5 min, while addition of ATP apparently accelerated the activation[24]. As the ATP-binding motif mapped in this study is highly conserved among species, ATP-dependent activation is likely a general mechanism for the bestrophin family of channels.

It should be noted that although the diminished affinities of the KpBest A2/L177T mutants and bBest2 I201T to ATP analogs strongly suggest a direct involvement of loop 2 on hBest1 (199–203) in ATP binding, a direct interaction between ATP and hBest1 has not been tested due to the lack of purified hBest1, while the structure of the ATP-binding pocket still remains elusive. We are actively exploring the purification of hBest1 channels and cocrystallization of ligand-bound KpBest in order to answer these important questions (Supplementary Table 3).

Extracellular ATP is a primary candidate substance of the light peak, an electrophysiological response in the eyes upon light exposure[25,26]. It has been proposed that ATP released by photoreceptors stimulates the increase of cytosolic $Ca^{2+}$ concentration in RPE through G protein-coupled purinergic receptors, subsequently generating a $Cl^-$ conductance at the basolateral membrane of RPE. Here, we discovered that intracellular ATP enhances the current amplitudes of both endogenous $Ca^{2+}$-dependent $Cl^-$ currents in WT iPSC-RPE and $Ca^{2+}$-dependent $Cl^-$ currents from exogenously expressed WT hBest1 in defective P274R iPSC-RPE. Although it is unclear if intracellular ATP in RPE fluctuates in response to light stimuli, activation of hBest1 through direct interaction provides a second mode of control for $Ca^{2+}$-dependent $Cl^-$ currents by ATP in human RPE. Interestingly, the aging of RPE cells is associated with a decrease of intracellular ATP levels[27], which may contribute to the progressive retinal phenotypes seen in some BEST1 patients.

KpBest and cBest1 structures are very similar: RMSDs are 2.4 Å in CCP4 superpose using the secondary structure matching mode (Fig. 7a). The two structures also have very similar pore radii at the conserved restrictions in the channel ion conducting pathway (Supplementary Table 2). On the other hand, hBest1 shares only 14% sequence identity with KpBest, much lower than that with cBest1 (74%, on residues 1–405). The unique combination of structural similarity and sequence divergence allows us to pinpoint critical residuals on the channel in a structure-based sequence alignment (Fig. 3a), as demonstrated by the mapping of the ATP-binding motif in this study. Therefore, KpBest is a very powerful tool for elucidating the function and structure of hBest1.

## Methods

**MST analysis.** Purified KpBest and bBest2 proteins were labeled with a labeling kit (Monolith NT RED NHS NT647) according to the manufacturer's instructions (NanoTemper Technologies). The molecule ratios of dye to protein were determined by absorbance at 650 and 280 nm. Buffer-exchange column chromatography was used to remove labeling reagents, and labeled proteins were eluted with MST reaction buffer: 40 mM HEPES (pH 7.8), 200 mM NaCl, 0.1 mM Tris [2-carbox-yethyl] phosphine (TCEP), and 0.05% n-dodecyl-β-D-maltopyranoside (DDM). The concentration of labeled proteins was adjusted to 5–20 nM for MST measurements. Up to 16 concentrations of twofold dilutions of unlabeled ligand were prepared and mixed with equal volume of labeled proteins followed by incubation

on ice for at least 10 min before MST measurement, which was performed on a Monolith 115 (NanoTemper) at room temperature. The power settings of light-emitting diode and MST were adjusted to 40% and 20%, respectively. Thermophoresis results were read out as normalized fluorescence ($F_{norm} = F_{hot}/F_{cold}$). As $F_{norm}$ varies between different labeling experiments, it was normalized as fraction bound to yield a binding curve (Supplementary Figures 1, 3, and 4), which was fitted to calculate the binding constants. The measured $K_d$s were equilibrium values, as the binding curves from samples (mix of labeled protein and non-fluorescent ligand) with different incubation times (10 or 40 min) stayed consistent. The results were analyzed with NanoTemper's NT analysis software. Ligands used in this study were sodium salts of ATP, ADP, AMP, and ATPγS (Sigma). The labeling procedure does not affect ATP binding, as the $EC_{50}$ of ATP in channel activation was unaffected with labeled proteins.

**Culture of iPSC-RPE.** iPSC-RPE cells were cultured in matrigel-coated dishes with RPE medium, and validated for RPE fate by well-established markers RPE65, Bestrophin1, CRALBP, MITF, and PAX6[16]. The I201T and P274R mutations in the patient-derived iPSC-RPEs were verified by sequencing[16]. All the iPSC-RPE cells were gifted from Stephen Tsang at Columbia University Medical Center and tested at their passage 1.

**Immunofluorescence.** iPSC-RPE cells were washed once with PBS and fixed in 4% paraformaldehyde at room temperature for 45 min. The fixed cells were washed with PBS twice and incubated in PBS with 2% donkey serum and 0.1% Triton X-100 for 45 min. Then, the samples were incubated with hBest1 antibody (1:200, Novus Biologicals, NB300-164) at room temperature for 2 h, followed by incubation with Alexa Fluor 555-conjugated IgG (1:1,000, Thermo Fisher Scientific, A-21422) at room temperature for 1 h. Immunofluorescent samples were analyzed by confocal microscopy (Nikon Ti Eclipse inverted microscope for scanning confocal microscopy, Japan).

**Cell lines.** HEK293 cells authenticated by short tandem repeat DNA profiling were kindly gifted from David Yule at University of Rochester. DMEM supplemented with 10% FBS and 100 µg ml$^{-1}$ penicillin–streptomycin was used for HEK293 cell culture. No mycoplasma contamination was found.

**Transfection.** Twenty to twenty-four hours before transfection, HEK293 cells were split into new 6-cm culture dishes at 50% confluency. The calcium phosphate precipitation method was used to transfect HEK293 cells with bBest2 WT or I201T (6 µg) and T antigen (2 µg). The transfection mix was removed after 4–8 h, and cells were washed with PBS and cultured in supplemented DMEM. Twenty-four hours after transfection, cells were split onto fibronectin-coated glass coverslips for patch clamp[28].

**Electrophysiology.** Electrophysiological analyses of RPE and HEK cells were conducted 24–72 h after cell split (with or without baculovirus infection) and transfection, respectively. Whole-cell patch clamp recording was performed with an EPC10 patch clamp amplifier (HEKA Electronics) controlled by Patchmaster (HEKA)[28]. Micropipettes were pulled and fashioned from filamented 1.5 mm thin-walled glass (WPI Instruments), and filled with internal solution containing (in mM): 110 CsCl, 10 EGTA, ATP (sodium salt, added fresh), 10 HEPES (pH 7.4). The desired free $Ca^{2+}$ concentration (maxchelator.stanford.edu/CaMgATPEGTA-TS.htm) and osmolarity were obtained by adding $CaCl_2$ and glucose, respectively. Series resistance was typically 1.5–2.5 MΩ, with no electronic series resistance compensation. The recipe of external solution was (in mM): 115 NaCl, 5 KCl, 2 $CaCl_2$, 1 $MgCl_2$, and 10 HEPES (pH 7.4). Traces were acquired at a repetition interval of 4 s[29]. Currents were sampled at 25 kHz and filtered at 5 or 10 kHz. I–V curves were generated from a group of step potentials (−100 to +100 mV from a holding potential of 0 mV).

A 3:1 mixture of phosphatidylethanolamine and phosphatidylcholine (Avanti Polar Lipids) dissolved in decane was used to paint planar lipid bilayers across a 200-µm hole in polysulfonate cups (Warner Instruments) separating two chambers: the trans chamber (representing the luminal compartment) was connected to the head stage input of a bilayer voltage clamp amplifier, while the cis chamber (representing the cytoplasmic compartment) was held at virtual ground. Components in the cis and trans solutions were (in mM): 150 NaCl and 10 HEPES (pH 7.4). ATP (sodium salt) was freshly added in the trans solution. In Supplementary Figure 1a, EGTA and/or $Ca^{2+}$ were supplemented as indicated. Purified proteins were added to the cis side and fused with the lipid bilayers. Currents from fused channels were recorded using a Bilayer Clamp BC-525D (Warner Instruments, LLC, CT), filtered at 1 kHz using a low-pass bessel filter 8 pole (Warner Instruments, LLC, CT), and digitized at 4 kHz. All experiments were conducted at room temperature (23 ± 2 °C). All statistical results were from three independent protein purifications.

**Baculovirus production.** BacMam baculoviruses bearing bBest2 (1–406)-GFP or hBest1 (WT/Mut)-GFP were generated[30]. In brief, BacMam plasmids containing the gene/fragment of interest were transformed into DH10Bac competent cells to

generate Bacmids, which were extracted to transfect insect SF9 cells (gifted from Ravi Kalathur) for baculovirus production. Baculoviruses were introduced into RPE culture 24 h after cell split (MOI = 100)[31].

**Molecular cloning**. The full-length bBest2 gene was synthesized by Genscript. A truncated bBest2 containing residues 1–406 was amplified by polymerase chain reaction, and was inserted into a BacMam mammalian expression vector. Point mutations of KpBest, bBest2, and hBest1 were made using the In-fusion Cloning Kit (Clontech). All constructs were verified by sequencing.

**Immunoprecipitation and immunoblotting**. RPE cells plated in 35-mm cell culture dishes were harvested 48 h after baculovirus infection[31]. Cells were washed once with PBS and lysed in 0.5 mL precooled lysis buffer (50 mmol L$^{-1}$ Tris-HCl, 150 mmol L$^{-1}$ NaCl, 1% NP-40) supplemented with protease inhibitors for 30 min at 4 °C. Cell lysates were centrifuged at 10,000×$g$ for 15 min at 4 °C. The supernatant was collected and incubated with 50 μL protein G beads slurry for 1 h. After centrifugation, the precleared supernatant was incubated with 4 μg GFP antibody (1:5000 Invitrogen, A6455) and 50 μL protein G slurry for 1 h at 4 °C on a rotator. The beads were collected after centrifugation and washed four times with lysis buffer. A 50 μL Laemmli buffer was used to resuspend the beads, followed by incubation at 95 °C for 5 min. The supernatants were collected for sodium dodecyl sulfate polyacrylamide gel electrophoresis and Western blot analyses. Immunoblotting was performed with hBest1 primary antibody (1:500 Novus Biologicals, NB300-164) and fluorophore-conjugated secondary antibody (1:10,000 Thermo-Fisher, SA5-35521), and subsequently detected by infrared imaging.

**Protein production and purification**. BL21 plysS cells were kindly gifted from Wayne Hendrickson. For KpBest production, BL21 plysS cells containing KpBest expression vectors were cultured overnight, inoculated 1:100 (v/v) into fresh TB media, and grown at 37 °C to OD 0.6–0.8. The culture was then induced with 0.4 mM IPTG and grown at 20 °C overnight. HEK293-F cells were gifted from Ravi Kalathur. To produce bBest2, HEK293-F cells were infected with BacMam baculoviruses bearing bBest2 (MOI = 5), and cultured at 37 °C for 72–96 h before harvesting[32]. 10 mM sodium butyrate was added to HEK293-F culture 24 h post infection.

Cells expressing targeted proteins were harvested by centrifugation at 4 °C, and stored at −80 °C until use. A common protein purification protocol was utilized for all KpBest and bBest2 proteins in this study[32]. In brief, cell pellets were resuspended in a buffer containing 50 mM HEPES (pH 7.8), 300 mM NaCl, 5% glycerol, 20 mM imidazole and 0.5 mM TCEP, and lysed using an emulsiflex-C3 high pressure homogenizer (Avestin). The cell lysate was incubated with a final concentration of 2% (w/v) DDM for 1 h at 20 °C. The nondissolved content was pelleted by ultracentrifugation at 150,000×$g$ for 30 min. The supernatant was carefully collected and loaded to a pre-equilibrated 5 mL HisTrap Ni$^{2+}$-NTA affinity column (GE Healthcare). After 13 column-volume buffer wash, the protein was eluted with a buffer containing 25 mM HEPES (pH 7.8), 200 mM NaCl, 5% glycerol, 500 mM imidazole, 0.1 mM TCEP and 0.05% (w/v) DDM. The 10× His (for KpBest) or GFP-10× His (for bBest2) tags were removed by incubating with TEV proteinase at 1:1 mass ratio at 4 °C for 30 min. The resulting samples were concentrated with 100 kDa centrifugal filter units (Amicon Ultra-15, Millipore) to a final volume of 400–500 μL for size exclusion chromatography with a Superdex-200 column by HPLC (AKTA pure 25, GE).

**Electrophysiological data and statistical analyses**. Whole-cell patch clamp data were processed off-line in Patchmaster. Statistical analyses were performed using built-in functions in Origin. We examined a sufficient number of samples to reach statistical conclusion according to the specific method utilized in that experiment. Statistically significant differences ($P < 0.05$) between means of two groups were determined by Student's $t$ test, while comparisons of more than two groups were performed by one-way ANOVA and Bonferroni post hoc analyses. Data are presented as means ± s.e.m.[33].

**Structure analysis**. Homology models for hBest1 were generated using MOD-ELLER[34]. All figures were made in PyMOL.

**Data availability**. Data supporting the findings of this manuscript are available from the corresponding author upon reasonable request.

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

## Acknowledgments

We thank Henry Colecraft and David Yule for comments on the paper, Austin Hopia-vuori for editing, Yota Fukuda for suggestions on model analysis, Stephen Tsang and Yao Li for iPSC-RPE cells, Wayne Hendrickson and Alexander Sobolevsky for help on mammalian bestrophin screen, and Seneca Hutson for help on making constructs. MST experiments were performed at the Rockefeller University (High-Throughput Screening and Spectroscopy Resource Center) and New York Structural Biology Center (Center on Membrane Protein Production and Analysis) (GM116799), and we thank Lavoisier S. Ramos-Espiritu, Carolina Adura Alcaino, J. Fraser Glickman and Ravi Kalathur for assistance and technical support. S.C. was supported by Sun Yat-Sen University "100 Top Talents Program (II)," Special fund for scientific and technological innovation strategy of Guangdong Province of China (2017A030313145) and the National Natural Science Foundation of China (Grant no. 31770801). This work was supported by NIH grants EY025290, GM127652, and University of Rochester start-up funding to T.Y.

## Author contributions

Y.Z. performed experiments, analyzed data, and wrote the paper; A.K. performed experiments; N.W. maintained RPE culture and made the viruses; C.J. performed experiments; S.C. analyzed structural data; T.Y. designed and performed experiments, analyzed data, made figures and wrote the paper.

## Additional information

**Competing interests:** The authors declare no competing interests.

