## [Peer Review File · Nature Communications]

Reviewers' comments:

Reviewer #1 (Remarks to the Author):

This manuscript from the Yang lab shows that bestrophins are regulated by ATP and that mutations in the putative ATP binding site are associated with human bestrophinopathy. These are novel results and the data generally support the authors' conclusions. A strength of the paper is the combination of proteins reconstituted into bilayers, whole-cell patch clamp of native currents in iPSC-derived RPE cells, and structural data. However, there are several weaknesses that need to be addressed.

The mechanism of stimulation of hBest1 current needs more documentation. (1) hBest1 current expressed in HEK cells runs down with time. Is the same true for iPSC-derived RPE cells? If so, is the effect of ATP explained by an effect on rundown? Please show data for the amplitude of hBest1 vs. time with and without ATP. (2) In cells with hBest1, is the effect of ATP explained by phosphorylation? Even though Mg is not added to the internal solution, there may be enough residual Mg present in the cells to support phosphorylation. Does AMPPNP, which does not support phosphorylation, also stimulate the current? (3) Is bBest2 stimulated by ATP? The authors use bBest2 to measure ATP binding to a mammalian bestrophin. The binding data should be accompanied by functional data.

The structural data provide a logical rationale for how ATP could regulate channel gating, but the impact is weakened somewhat by the use of KpBest. Why not do these experiments with bBest2 or cBest1 which have higher sequence identity to hBest1? In any case, the authors propose that loop 2 is involved in ATP binding, but the data supporting this contention are not as strong as I would like to see. I was hoping that a crystal structure of KpBest with ATP bound would be presented. Alternatively, an in silico docking experiment or molecular dynamics simulation could provide additional support. The authors suggest the two Arg residues are key for ATP binding, but KpBest does not have these two Rs, but one K. A more detailed mutagenesis of this region is required. For example, does substituting KpBest loop 2 into hBest1 retain ATP binding? As it stands, the KpBest1 I180A mutant seems a little tangential to the key issue at hand.

Minor issues.

P3. "These functional results suggest that bestrophins have additional activator(s) besides Ca²⁺" This statement does not seem to follow logically from the preceding narrative. Why does the Ca-independent activation of some bestrophins and the high sensitivity of others to Ca suggest there are other activators? Maybe the channels are always open physiologically? On the other hand, the next paragraph makes a more cogent argument for the likelihood of another regulator.

P4. "Using KpBest as a search tool, we mapped a critical ATP-binding motif adjacent to a conserved activation gate in the channel ion conducting pathway." I did not understand what this means. After finishing reading the paper, I get the idea, but actually the sequence/structure of KpBest2 itself provided little or no insight into the location of the ATP binding site.

P4. "A crucial clue on" – "on" should be "about" or "to", I think.

Fig. 1C x-axis label "ATP analog" is misleading as the black curve is ATP itself, not an analog.

Page 4, the authors state, but do not show data, that KpBest is not activated by Ca in bilayers. These data should be shown. Also, the solution used in the bilayer experiments does not contain EGTA. How can the authors be certain that the ATP does not contain contaminating Ca? The authors should repeat these experiments with EGTA.

The authors should show an SDS-PAGE gel of the KpBest preparation to validate their statement

that "no other protein" was present.

The data suggest that the channel is gated by the free acid of ATP (without complexed Mg). It would be interesting to know whether Mg.ATP also works.

Fig. 3A legend should state which regions are presumed ATP binding sites. It was not clear from the figure until I noticed the miniscule labels for loops 1-4.

Fig. 4C. states $n=3$. I presume this means 3 bilayers for each construct, or does it mean 3 different protein purifications? I think a larger n is required from at least 2 different protein purifications. Also, P_o was calculated from how many events? All-points histograms for the entire experiment should be shown accompanying the sample traces.

Reviewer #2 (Remarks to the Author):

NCOMMS-18-08461-T

Zhang et al. report the identification of ATP as a co-regulator of the calcium activated chloride channel Bestrophin (BEST) using a combination of binding assays and functional studies. The data support this main conclusion.

The authors claim on p. 8 that the A1 and A3 mutants of KpBest lost interaction with ATP because of non-specific channel disruption. However, there are no data presented to support this claim. The lack of data is surprising, as it seems that these proteins were purified for the experiments. Thus, the authors must have size exclusion data that could be presented and that would establish whether the gross properties of these mutants are similar to or different from the wild-type channel. Such data would help establish whether disruption of the putative ATP also destroys the integrity of the channel and should be shown.

The authors perform functional experiments in the background of cells having a non-functional mutation, P274R. Because Best channels are multimers, this background raises the possibility that the measurements they make are do not purely represent the introduced channels but may include mixed heteromultimers containing the introduced channel and some number of P274R mutants. Even though P274R is non-functional on its own, it is unclear whether one or a few of these mutant subunits could form a functional channel when co-assembled with other functional subunits. The authors need to address this point as otherwise the exact nature of the measured channels in Fig. 5 is unclear making it difficult to draw any conclusions from these experiments.

The authors determine the structure of KpBEST mutant, I180A and claim (p.12) that this mutant dramatically increases the activation gate opening, but does not affect the neck. No evidence for the latter claim is presented. Please show this point.

The authors claim that KpBEST and cBest1 have a 'very similar' all atom RMSD. The value is 12.3Å!!! and is reduced after 5 cycles (of what?) to 4.5Å Neither value can be used to claim similarity. Some explanation is needed. Inspection of Fig. 7A indicates that at least two of the transmembrane helices appear to have a register shift. This may be the source of the mismatch. Further, given that the sequences of the two proteins are not identical, the authors might be better served focusing on the Calpha superposition. The figure clearly indicates some similarity, and I suspect that the issues with the large numbers are due to sidechain mismatches and the two (or maybe three, it is hard to see from the figure, especially as the blue and green are similar hues)

helix mismatches.

Reference to Fig. S3 should occur immediately after the claim that the authors obtained the first purified hBest1 protein (Line 34 of last full paragraph on p. 6).

Fig. 7C and d would be improved if the two structures each had labeling indicating which is KpBest and which is cBEST1.

Reviewer #3 (Remarks to the Author):

The manuscript by Zhang et al reported activation of a bacterial homolog of human bestrophin channel by ATP and its analogs. The authors went on to show that L177T can abolish ATP binding and also renders the bacterial channel insensitive to ATP. The simplest explanation is that ATP binds to KpBest through L177, although an allosteric mechanism cannot be ruled out.

The authors then expanded the study to human and bovine bestrophin channels, and eventually concluded that the mammalian channel can also bind to and be activated by ATP and that one of the intracellular loop is likely the binding site for ATP.

While I appreciate the combination of structural and functional approaches, and recognize that the use of induced human RPE cells for functional studies could add significance and physiological relevance to the study, I feel that several key experiments are missing and therefore the conclusions on the mammalian channels are not well justified. In the current format, the manuscript only demonstrated that KpBest is activated by ATP, and that ATP activates KpBest through changes at regions near L177.

Here are the questions I have:

1. Studies of ATP activation of bovine or human bestrophin channel should be done on either purified protein reconstituted into liposomes (for the bovine channel) or on a heterologous expression system such as HEK cells (for either bovine or human channels). That way, bestrophin channel currents can be rigorously validated. This step is necessary to establish that ATP activates mammalian bestrophin channels.
2. Once step 1 is complete, mutations can then be tested to identify regions that are sensitive to ATP activation. At this point, ATP binding to either purified bovine or chicken bestrophin (both the wild type and mutations at the homologous position of L177) should be measured, and functions of the wild type and mutant channels recorded and compared.
3. Once step 1 and 2 are complete, the recordings on induced human RPE cells would then become impactful. And even at this stage, validation of the recorded currents on RPE cells is necessary because there are other channels that could produce chloride current on a native cell.
4. Independent of issues with the mammalian bestrophin channels, the logic of presenting the structure of the bacterial I180A channel is not clear. This is a channel that has a high Popen without the presence of ATP. The opening of the channel is almost entirely due to the truncation of the Ile side chain. In which way does it represent an ATP activated channel? Does it still bind to ATP? If so, how does channel activity change in the presence of ATP?
5. If ATP affinity is at the micromolar range, a structure of ATP in complex with KpBest should be attainable and that would address the question of whether ATP activates the channel by directly interacting with L177 or through an allosteric effect.
6. Related to #5, the loop mutations that produced functionally null channels should be examined for ATP binding. This should be done for both the BpBest and mammalian bestrophins. The reason given for not following up on these mutations is not compelling.

We thank the reviewers for their supportive comments on the manuscript and for their constructive critique of the work. We have prepared a revised version of our paper that incorporates changes to address the major questions and reviewer comments. We copy the referee comments below verbatim and respond to each of the points, describing how we have modified the paper to address the concerns.

Reviewer #1

The mechanism of stimulation of hBest1 current needs more documentation.

1. *“hBest1 current expressed in HEK cells runs down with time. Is the same true for iPSC-derived RPE cells? If so, is the effect of ATP explained by an effect on rundown? Please show data for the amplitude of hBest1 vs. time with and without ATP.”*

Response: iPSC-derived RPE cells have run up with medium $[Ca^{2+}]_i$. We have added current amplitudes vs. time in the absence and presence of NaATP in **Supplementary Fig. 2c**. The different electrophysiological behaviors of hBest1 in HEK cells vs. in iPSC-derived RPE cells may reflect the variances between the host cell types.

2. *“In cells with hBest1, is the effect of ATP explained by phosphorylation? Even though Mg is not added to the internal solution, there may be enough residual Mg present in the cells to support phosphorylation. Does AMPPNP, which does not support phosphorylation, also stimulate the current?”*

Response: The reviewer raised a very good point. To address this question, we examined the stimulatory effect of ATP γ S, which does not support phosphorylation, instead of AMPPNP for the consistency of the paper. As shown in **Fig. 2d**, ATP γ S enhances the amplitude of Ca^{2+} -dependent Cl^- current in human RPE cells to a similar level as NaATP, suggesting that the observed ATP-dependent stimulation does not require phosphorylation.

3. *“Is bBest2 stimulated by ATP? The authors use bBest2 to measure ATP binding to a mammalian bestrophin. The binding data should be accompanied by functional data.”*

Response: We measured the current amplitudes of bBest2 in HEK293 cells with or without ATP. As shown in **Fig. 3d**, the activity of bBest2 is stimulated by ATP.

4. *“The structural data provide a logical rationale for how ATP could regulate channel gating, but the impact is weakened somewhat by the use of KpBest. Why not do these experiments with bBest2 or cBest1 which have higher sequence identity to hBest1?”*

Response: We have been actively working on solving the structure of bBest2, but have not obtained a structure yet. We do not have the capacity to work on cBest1.

“...In any case, the authors propose that loop 2 is involved in ATP binding, but the data supporting this contention are not as strong as I would like to see. I was hoping that a crystal structure of KpBest with ATP bound would be presented. Alternatively, an in silico docking experiment or molecular dynamics simulation could provide additional support.”

Response: We have been actively working on the co-crystal structures of bBest2-ATP and KpBest-ATP, but have not succeeded in solving them yet (added in **Discussion** on Page 14, Paragraph 2). We attempted docking with AutoDock and ZDOCK, but didn't find any reasonable ATP binding site, suggesting that ATP binding is associated with a dramatic conformational

change in the structure. This idea is also supported by crystal soaking experiments, in which well-grown KpBest crystals were incubated with ATP/ATP γ S before cryo-preservation, as the diffraction resolution of the crystals was significantly impaired even after half an hour soaking, decreasing from ~3 Å to over 10 Å.

“...The authors suggest the two Arg residues are key for ATP binding, but KpBest does not have these two Rs, but one K. A more detailed mutagenesis of this region is required. For example, does substituting KpBest loop 2 into hBest1 retain ATP binding?”

Response: The two R residues are conserved in mammalian and chicken bestrophins, while in KpBest, K176 in loop2 and R172 next to loop2 may be involved in ATP binding. As the sequences of KpBest and hBest1 are quite different, we respectfully think that the loop swapping experiments could be complicated and difficult to interpret. Nevertheless, the reviewer’s point is very well taken, and we will further investigate the ATP binding site in follow-up studies.

“...As it stands, the KpBest1 I180A mutant seems a little tangential to the key issue at hand.”

Response: We have removed the KpBest I180A mutant data as suggested.

5. *“P3. ‘These functional results suggest that bestrophins have additional activator(s) besides Ca²⁺’ This statement does not seem to follow logically from the preceding narrative. Why does the Ca-independent activation of some bestrophins and the high sensitivity of others to Ca suggest there are other activators? Maybe the channels are always open physiologically? On the other hand, the next paragraph makes a more cogent argument for the likelihood of another regulator.”*

Response: As the reviewer suggested, we have added the possibility that bestrophins may remain constantly open in the last sentence of the referred paragraph:

“These functional results suggest that bestrophins **either remain constantly open or** have additional activator(s) besides Ca²⁺ under physiological conditions.”

6. *“P4. ‘Using KpBest as a search tool, we mapped a critical ATP-binding motif adjacent to a conserved activation gate in the channel ion conducting pathway.’ I did not understand what this means. After finishing reading the paper, I get the idea, but actually the sequence/structure of KpBest2 itself provided little or no insight into the location of the ATP binding site.”*

Response: The reviewer’s comment is very well taken. We have revised the sentence to **“Using purified KpBest mutant proteins as a probing tool**, we mapped a critical ATP-binding motif adjacent to a conserved activation gate in the channel ion conducting pathway.”

7. *“P4. ‘A crucial clue on’ – ‘on’ should be ‘about’ or ‘to’, I think.”*

Response: We have replaced “on” with “about” as the reviewer suggested.

8. *“Fig. 1C x-axis label ‘ATP analog’ is misleading as the black curve is ATP itself, not an analog.”*

Response: We have relabeled Fig. 1C x-axis as “Ligand”.

9. *“Page 4, the authors state, but do not show data, that KpBest is not activated by Ca in bilayers. These data should be shown. Also, the solution used in the bilayer experiments does not contain*

EGTA. How can the authors be certain that the ATP does not contain contaminating Ca? The authors should repeat these experiments with EGTA”.

Response: As the reviewer suggested, we have added the influence of Ca²⁺ and ATP on KpBest in the presence of EGTA. As shown in **Supplementary Fig. 1a**, the open probability of KpBest was significantly stimulated by 2 mM ATP, but not responsive to 1 μM free Ca²⁺, indicating that KpBest is not activated by Ca²⁺. Moreover, 2 mM ATP activates KpBest to a similar level with or without EGTA in the bilayer solution (compare Fig. 1b and **Supplementary Fig. 1a**), suggesting that contamination of Ca²⁺ in ATP is negligible.

10. “The authors should show an SDS-PAGE gel of the KpBest preparation to validate their statement that “no other protein” was present”.

Response: As the reviewer suggested, size exclusion profile and SDS-PAGE of purified KpBest protein have been added in **Supplementary Fig. 1b-c**.

11. “The data suggest that the channel is gated by the free acid of ATP (without complexed Mg). It would be interesting to know whether Mg.ATP also works.”

Response: We agree with the reviewer, and are actively testing the influence of MgATP on bestrophins in a follow-up study.

12. “Fig. 3A legend should state which regions are presumed ATP binding sites. It was not clear from the figure until I noticed the miniscule labels for loops 1-4.”

Response: We have added the description of loops 1-4 in Fig. 3A legend according to the reviewer’s advice.

13. “Fig. 4C. states n=3. I presume this means 3 bilayers for each construct, or does it mean 3 different protein purifications? I think a larger n is required from at least 2 different protein purifications. Also, Po was calculated from how many events? All-points histograms for the entire experiment should be shown accompanying the sample traces.”

Response: We performed bilayers from 3 different protein purifications. We have added the information in **Methods**. The histograms for the A2 (loop 2) and A4 (loop 4) mutants have been added in **Supplementary Fig. 4c-d**. A1 (loop 1) and A3 (loop 3) mutants were non-functional in bilayer.

Reviewer #2

1. “The authors claim on p. 8 that the A1 and A3 mutants of KpBest lost interaction with ATP because of non-specific channel disruption. However, there are no data presented to support this claim. The lack of data is surprising, as it seems that these proteins were purified for the experiments. Thus, the authors must have size exclusion data that could be presented and that would establish whether the gross properties of these mutants are similar to or different from the wild-type channel. Such data would help establish whether disruption of the putative ATP also destroys the integrity of the channel and should be shown”.

Response: As the reviewer suggested, we now have added the MST data for A1/A3 and size exclusion results for A1-A4 in **Supplementary Fig. 4**. The size exclusion profiles of A1-A4 are indistinguishable from that of the WT KpBest (Supplementary Fig. 1), suggesting that the integrity

of the channel is retained in all these mutants. We agree with the reviewer that the involvement of loops 1 and 3 in ATP binding cannot be ruled out. However, it is difficult to further explore this possibility when conserved residues in these motifs are irreplaceable for constituting a functional channel. It should be noted that in the literature mutagenesis of hBest1 often results in loss of channel function, stressing the structural rigidity of the channel. Therefore, it is more informative to focus on loop 2 which is specifically involved in ATP binding and ATP-dependent activation. We have revised the text on Page 8 accordingly.

2. *“The authors perform functional experiments in the background of cells having a non-functional mutation, P274R. Because Best channels are multimers, this background raises the possibility that the measurements they make do not purely represent the introduced channels but may include mixed heteromultimers containing the introduced channel and some number of P274R mutants. Even though P274R is non-functional on its own, it is unclear whether one or a few of these mutant subunits could form a functional channel when co-assembled with other functional subunits. The authors need to address this point as otherwise the exact nature of the measured channels in Fig. 5 is unclear making it difficult to draw any conclusions from these experiments.”*

Response: We have added co-IP results in **Supplementary Fig. 5**, showing that the P274R mutant does not interact with WT or any of the A1-A4 mutants. Therefore, P274R cannot interfere with the function of the introduced WT or A1-A4 channels by forming heteromultimers.

3. *“The authors determine the structure of KpBEST mutant, I180A and claim (p.12) that this mutant dramatically increases the activation gate opening, but does not affect the neck. No evidence for the latter claim is presented. Please show this point.”*

Response: We have removed the I180A data as suggested by the other two reviewers.

4. *“The authors claim that KpBEST and cBest1 have a ‘very similar’ all atom RMSD. The value is 12.3Å!!! and is reduced after 5 cycles (of what?) to 4.5Å Neither value can be used to claim similarity. Some explanation is needed. Inspection of Fig. 7A indicates that at least two of the transmembrane helices appear to have a register shift. This may be the source of the mismatch. Further, given that the sequences of the two proteins are not identical, the authors might be better served focusing on the Alpha superposition. The figure clearly indicates some similarity, and I suspect that the issues with the large numbers are due to sidechain mismatches and the two (or maybe three, it is hard to see from the figure, especially as the blue and green are similar hues) helix mismatches.”*

Response: The reviewer raised an important point, and we agree that the Pymol alignment results were not clear. In order to re-evaluate the structure similarity between KpBest and cBest1, we performed CCP4 superpose using the Secondary Structure Matching mode, and the resulting RMSD between KpBest (4WD8) and cBest1 (4RDQ, without antibody molecules) is 2.4216 Å (1077 residues alignment). Thus, our new alignment data support that the overall structures of KpBest and cBest are very similar. We have revised the manuscript accordingly (Page 15, paragraph 2).

5. *“Reference to Fig. S3 should occur immediately after the claim that the authors obtained the first purified bBest2 protein (Line 34 of last full paragraph on p. 6).”*

Response: We have revised the reference as the reviewer suggested.

6. "Fig. 7C and d would be improved if the two structures each had labeling indicating which is KpBest and which is cBEST1."

Response: We have labelled KpBest and cBest1 in Fig. 7c-d as the reviewer suggested.

Reviewer #3

While I appreciate the combination of structural and functional approaches, and recognize that the use of induced human RPE cells for functional studies could add significance and physiological relevance to the study, I feel that several key experiments are missing and therefore the conclusions on the mammalian channels are not well justified. In the current format, the manuscript only demonstrated that KpBest is activated by ATP, and that ATP activates KpBest through changes at regions near L177.

Here are the questions I have:

1. "Studies of ATP activation of bovine or human bestrophin channel should be done on either purified protein reconstituted into liposomes (for the bovine channel) or on a heterologous expression system such as HEK cells (for either bovine or human channels). That way, bestrophin channel currents can be rigorously validated. This step is necessary to establish that ATP activates mammalian bestrophin channels."

Response: As the reviewer suggested, we have measured the current amplitudes of bBest2 expressed in HEK293 cells with or without ATP. As shown in **Fig. 3d**, the activity of bBest2 is significantly stimulated by ATP.

2. "Once step 1 is complete, mutations can then be tested to identify regions that are sensitive to ATP activation. At this point, ATP binding to either purified bovine or chicken bestrophin (both the wild type and mutations at the homologous position of L177) should be measured, and functions of the wild type and mutant channels recorded and compared."

Response: We have purified the corresponding bBest2-I201T mutant protein for MST measurement and found that ATP binding is disrupted, as shown in **Supplementary Fig. 3d**. Consistently, the channel activity of bBest2-I201T expressed in HEK293 cells is not responsive to ATP, as shown in **Supplementary Fig. 3c**. These new bBest2-I201T results are in sharp contrast to those from WT bBest2 (Fig. 3b-3d), providing strong evidence for our conclusion that the patient-derived hBest1-I201T mutation causes deficiencies in ATP binding and ATP-dependent activation of the channel.

3. "Once step 1 and 2 are complete, the recordings on induced human RPE cells would then become impactful. And even at this stage, validation of the recorded currents on RPE cells is necessary because there are other channels that could produce chloride current on a native cell."

Response: We have recently shown that hBest1 is indispensable for and dictates Ca²⁺-dependent Cl⁻ current in human RPE cells (Li et al. 2017 *eLife*).

4. "Independent of issues with the mammalian bestrophin channels, the logic of presenting the structure of the bacterial I180A channel is not clear. This is a channel that has a high Popen without the presence of ATP. The opening of the channel is almost entirely due to the truncation

of the Ile side chain. In which way does it represent an ATP activated channel? Does it still bind to ATP? If so, how does channel activity change in the presence of ATP?"

Response: We appreciate the reviewer's concerns and have removed the I180A data from the manuscript.

5. "If ATP affinity is at the micromolar range, a structure of ATP in complex with KpBest should be attainable and that would address the question of whether ATP activates the channel by directly interacting with L177 or through an allosteric effect."

Response: We are actively working on the co-crystal structures of bBest2-ATP and KpBest-ATP, but have not succeeded in solving them yet (added in **Discussion** on Page 14, Paragraph 2). Please also refer to our response to Reviewer 1's question #4.

6. "Related to #5, the loop mutations that produced functionally null channels should be examined for ATP binding. This should be done for both the KpBest and mammalian bestrophins. The reason given for not following up on these mutations is not compelling."

Response: We have now added MST data for KpBest A1 and A3 in **Supplementary Fig. 4f**, showing no significant ATP binding to either mutant. We speculate that bBest2 A1 and A3 mutants are defective in ATP binding as well, but do not have enough capacity to purify them for MST experiments within a reasonable time frame. As an alternative, we have shown that bBest2 I201T (a mutation in loop 2) specifically lost ATP binding and ATP-dependent activation (**Supplementary Fig. 3c-d**), strongly supporting our main conclusion that loop 2 is critically involved in ATP binding. Please also refer to our response to Reviewer 2's question #1.

REVIEWERS' COMMENTS:

Reviewer #2 (Remarks to the Author):

The authors have addressed the points raised previously. The current manuscript is acceptable for publication pending edits.

One minor issue is that the new Figure S5 'exogenous' should be corrected to 'exogenous'. It would also help to label the endogenous hBest1 as 'endogenous hBest1 P247R' as the way the figure is labeled it looks as though the endogenous protein is the wild-type.

Reviewer #3 (Remarks to the Author):

I have no further comments.

Reviewer #2:

“The authors have addressed the points raised previously. The current manuscript is acceptable for publication pending edits.

One minor issue is that the new Figure S5 'exogenous' should be corrected to 'exogenous'. It would also help to label the endogenous hBest1 as 'endogenous hBest1 P247R' as the way the figure is labeled it looks as though the endogenous protein is the wild-type.”

Response: We appreciate the reviewer's comments and have relabeled Figure S5 as suggested.